# Capturing Actionable Dynamics
# with Structured Latent Ordinary Differential Equations

**Paidamoyo Chapfuwa**[1]   **Sherri Rose**[1]   **Lawrence Carin**[2]   **Edward Meeds**[3]   **Ricardo Henao**[4]

[1]Stanford University, USA
[2]KAUST, Saudi Arabia
[3]Microsoft Research, Cambridge, UK
[4]Duke University, USA

## Abstract

End-to-end learning of dynamical systems with black-box models, such as neural ordinary differential equations (ODEs), provides a flexible framework for learning dynamics from data without prescribing a mathematical model for the dynamics. Unfortunately, this flexibility comes at the cost of understanding the dynamical system, for which ODEs are used ubiquitously. Further, experimental data are collected under various conditions (inputs), such as treatments, or grouped in some way, such as part of sub-populations. Understanding the effects of these system inputs on system outputs is crucial to have any meaningful model of a dynamical system. To that end, we propose a structured latent ODE model that explicitly captures *system input* variations within its latent representation. Building on a static latent variable specification, our model learns (independent) stochastic factors of variation for each input to the system, thus separating the effects of the system inputs in the latent space. This approach provides actionable modeling through the *controlled generation* of time-series data for novel input combinations (or perturbations). Additionally, we propose a flexible approach for quantifying uncertainties, leveraging a quantile regression formulation. Results on challenging biological datasets show consistent improvements over competitive baselines in the controlled generation of observational data and inference of biologically meaningful system inputs.

## 1 INTRODUCTION

Dynamical systems are fundamental models in many scientific domains. Examples include the study of biological processes such as gene regulation (Calderhead et al., 2009), human cardiovascular systems (Zenker et al., 2007), epidemiology (Siettos and Russo, 2013), and synthetic biology (Roeder et al., 2019). The evolution of continuous-time dynamical systems are commonly modeled mathematically by ordinary differential equations (ODEs) as

$$\frac{d\boldsymbol{x}}{dt} = f\left(\boldsymbol{x}(t), t, \boldsymbol{u}(t)\right), \quad \boldsymbol{x}(0) = \boldsymbol{x}_0, \quad t \in [0, T], \quad (1)$$

and are governed by mathematical rules known as *dynamics* $f(\cdot)$, where $\boldsymbol{x}(t) \in \mathbb{R}^D$ is the *state* (snapshot of the process at time $t$) or solution of the ODE system, and $\boldsymbol{u}(t)$ are the system *inputs*. Moreover, given a state $\boldsymbol{x}_0$ as the *initial condition*, the dynamics define a temporal *trajectory* from a starting point at $t = 0$. Such systems can be categorized as deterministic *vs.* stochastic, or linear *vs.* nonlinear. In practice, we are given a set of noisy observations $\boldsymbol{y}(t) = m(t, \boldsymbol{x}(t))$ at $t = t_0, \ldots, t_T$, where $m(\cdot)$ is the unknown *emission* function, and we typically make assumptions to estimate functions $\{f(\cdot), \boldsymbol{x}(t), m(\cdot)\}$ parametrically or nonparametrically.

Classical *state-space* models, such as the Kalman filter (Kalman, 1960), assume a parametric *linear Gaussian state-space model* for the dynamics and emission functions. Because these assumptions are violated in practice and limit model flexibility, modifications were introduced which can generalize to nonlinear systems (Julier and Uhlmann, 1997, 2004). Recent variants of the *Gaussian state-space model* retain the Markovian structure of hidden Markov models and leverage neural networks for learning nonlinear dynamics and emission functions (Krishnan et al., 2017; Fraccaro et al., 2017; Miladinović et al., 2019).

While nonlinear systems are flexible, they are difficult to solve and rarely yield closed-form solutions for $\boldsymbol{x}(t)$. Hence, *implicit* approximations to numerical integration of system dynamics have been considered, *e.g.*, methods that directly solve for $\boldsymbol{x}(t)$ for a known $f(\cdot)$, leveraging the adaptive Euler method (Runge, 1895; Kutta, 1901; Alexander, 1990). Such approaches are computationally imprecise and challenging to scale for complex systems. Several ap-

*Accepted for the 38th Conference on Uncertainty in Artificial Intelligence* (UAI 2022).

proaches adopt gradient matching using Gaussian processes (GPs) (Calderhead et al., 2009; Graepel, 2003; Rasmussen, 2003), and related approaches based on a reproducing kernel Hilbert space (RKHS) (González et al., 2014) primarily, to avoid numerical integration. Unfortunately, kernel learning with GPs or RKHS is challenging to scale for large datasets and requires complete observability of $\boldsymbol{x}(t)$ (Ghosh et al., 2021). Alternatively, some methods conveniently presume discrete-time nonlinear dynamical modeling for deterministic and easy-to-evaluate state-space solutions, such as recurrent (or autoregressive) neural networks (Valpola and Karhunen, 2002; Karl et al., 2017; Yingzhen and Mandt, 2018), albeit constrained to pre-specified time-horizons.

We further divide methods that learn nonlinear dynamics according to assumptions required for estimating ODE dynamics, where $f(\cdot)$ is modeled as a neural network (Chen et al., 2018), or more recently, parameterized by a latent variable model (Rubanova et al., 2019), that leverages amortized variational inference (Kingma and Welling, 2013; Rezende et al., 2014). While a large body of machine learning approaches assume a known parametric form of the dynamics $f(\cdot)$ (Linial et al., 2021; Wan et al., 2001; Wenk et al., 2020), alternative flexible approaches assume that the parametric form of $f(\cdot)$ is unknown (Rubanova et al., 2019; Roeder et al., 2019). Moreover, several specifications of variational inference for latent variable *state-space* models have been proposed (Linial et al., 2021; Rubanova et al., 2019; Karl et al., 2017; Roeder et al., 2019; Miladinović et al., 2019; Yingzhen and Mandt, 2018; Fraccaro et al., 2017). Of these, only Roeder et al. (2019) considers a structured hierarchical latent variable model accounting for both observations and system inputs. So motivated, we adopt a data-driven approach to learn unknown functions $\{f(\cdot), m(\cdot)\}$ parameterized by neural networks. Moreover, we leverage a variational inference approach to learn a structured latent variable model (separating *input-* from *noise-specific* components) given observations $\boldsymbol{y}(t)$, as well as *static* system inputs $\boldsymbol{u}$, to characterize the unknown dynamics and emission functions.

Closely related to our work are latent variable state-space models focused on features that separate *static from dynamic* (Yingzhen and Mandt, 2018; Fraccaro et al., 2017), *domain-invariant from domain-specific* (Miladinović et al., 2019), *position from momentum* (Yildiz et al., 2019), and parameter (system input) estimation (Linial et al., 2021). In contrast, our work focuses on synthesizing observational data $\boldsymbol{y}(t)$ from dynamical systems given: $(i)$ combinations of previously unseen inputs $\boldsymbol{u}$ (also known as *zero-shot* learning), and $(ii)$ a simulated *continuous-time state-space* $\boldsymbol{x}(t)$ from an ODE solver. Controlled generation of observations under combinations of *system input* is foundational in experimental science for a mechanistic understanding of biology phenomena (Roeder et al., 2019; Yuan et al., 2021), particularly in scenarios when obtaining experimental data is expensive. Unlike Roeder et al. (2019), we do not impose

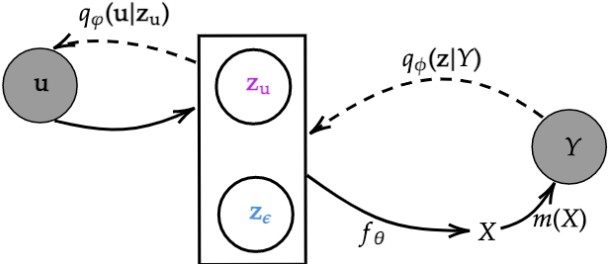

Figure 1: Illustration of the proposed structured latent ODE (SL-ODE) model. Generative: prior $\boldsymbol{z} = \{\boldsymbol{z_u}, \boldsymbol{z_\epsilon}\}$ (2) is mapped to states $X$ simulated from an ODE solver given *dynamics* $f_{\boldsymbol{\theta}}$ (4) to generate observations (*system outputs*) $Y$ from the *emission* function $m(\cdot)$. Inference: posterior $q_{\boldsymbol{\varphi},\phi}(\boldsymbol{z}|Y,\boldsymbol{u})$ is decomposed according to $q_\phi(\boldsymbol{z}|Y)$ and $q_{\boldsymbol{\varphi}}(\boldsymbol{u}|\boldsymbol{z_u})$ (9) where $\boldsymbol{u}$ are *system inputs*.

a hierarchical-latent structure or assume a known Gaussian emission process. Additionally our model enables inference of the system inputs $\boldsymbol{u}$ given observational data $\boldsymbol{y}(t)$, which is not considered in Roeder et al. (2019).

The key contributions of this paper are as follows:

- We present a principled statistical framework for integrating structured representation learning from *systems inputs* and *observations* with mechanistic models.
- We demonstrate that the proposed generative model accurately simulates system outputs (observations) given novel combinations or perturbations of system inputs, *i.e.*, zero-shot learning.
- We formulate a flexible quantile regression approach for quantifying uncertainties in generated observations.
- We demonstrate the benefits of integrating a structured latent ODE with a flexible emission function for improved performance over competitive baselines given challenging biological data: $(i)$ accurately inferring unknown *static* system inputs $\boldsymbol{u}$ from noisy observations $\boldsymbol{y}(t)$, and $(ii)$ improved uncertainty estimates of *observational* noise.

## 2 STRUCTURED LATENT ODE MODEL (SL-ODE)

We propose a mechanistic approach for generating observations governed by nonlinear dynamical systems. Figure 1 illustrates the proposed approach. Specifically, we leverage an amortized inference framework (Kingma and Welling, 2013; Rezende et al., 2014) to learn a structured latent representation given time-series observational data and static system inputs. Below we present the proposed generative process, including a quantile regression formulation for flexible (asymmetric) uncertainty estimation.

## 2.1 GENERATIVE PROCESS

We assume observations $\mathcal{D} = \{Y, \boldsymbol{u}\}_{i=1}^N$, where $Y_i \in \mathbb{R}^{K \times T}$ is a matrix of $K$ measurements at $T$ time points, for $i = 1, \ldots, N$ observations and $\boldsymbol{u}$ are the (auxiliary) *static* inputs (or system conditions). We propose a generative process that synthesizes $Y$ given $\boldsymbol{u}$ as follows

$$\boldsymbol{z_u} \sim p_\psi(\boldsymbol{z_u} \,|\, \boldsymbol{u}), \quad \boldsymbol{z_\epsilon} \sim p(\boldsymbol{z_\epsilon}), \quad \boldsymbol{z} = \{\boldsymbol{z_u}, \boldsymbol{z_\epsilon}\} \quad (2)$$

$$\frac{d\boldsymbol{x}}{dt} = f_{\boldsymbol{\theta}}(\boldsymbol{x}; \boldsymbol{z}, t) \quad (3)$$

$$X = \text{ODESolve}\,(f_{\boldsymbol{\theta}}, \boldsymbol{x}_0, (t_0, t_1, .., t_T)) \quad (4)$$

$$Y \sim p\,(Y | m_{\boldsymbol{\gamma}}(X), \sigma, \tau)\,, \quad (5)$$

where the functions defining $f_{\boldsymbol{\theta}}(\cdot)$, $p_\psi(\cdot)$, and $m_{\boldsymbol{\gamma}}(\cdot)$ are specified as neural networks parameterized by $\{\boldsymbol{\theta}, \psi, \boldsymbol{\gamma}\}$, respectively. We synthesize $Y$ in (5) as governed by black-box dynamics $f_{\boldsymbol{\theta}}(\cdot)$ in (3) parameterized by the latent representation $\boldsymbol{z}$ (composed of system inputs and process-noise) in (2). Moreover, the ODE solver (ODESolve) in (4) enables recovery of the *state-time* matrix $X$ at $\{t_0, \ldots, t_T\}$ for the corresponding observations $Y$. See Supplementary Material (SM) for detailed formulation of $f_{\boldsymbol{\theta}}(\cdot)$ and initial state mapping $\boldsymbol{z} \to \boldsymbol{x}_0$.

**Structured Latent-Space Representations** To enable controlled generation of system outputs (observations) from novel combinations or perturbations of system inputs, we specify a conditional prior that captures the relationships among heterogeneous system input values. We assign latent variable $\boldsymbol{z}$ to be the concatenation of *input-specific* $\boldsymbol{z_u}$ and *noise-specific* $\boldsymbol{z_\epsilon}$, variables with prior distributions $p_\psi(\boldsymbol{z_u}|\boldsymbol{u})$ and $p(\boldsymbol{z_\epsilon})$, respectively. Moreover, we learn a continuous and smooth representation of the input data in (2). We conveniently assume a Gaussian distribution:

$$p_\psi(\boldsymbol{z_u} \,|\, \boldsymbol{u}) = N\left(\boldsymbol{\mu}_\psi(\boldsymbol{u}), \text{diag}\left(\boldsymbol{\sigma}_\psi^2(\boldsymbol{u})\right)\right)\,, \quad (6)$$

where $\boldsymbol{\mu}_\psi(\cdot)$ and $\boldsymbol{\sigma}_\psi^2(\cdot)$ are the mean and variance functions of $\boldsymbol{u}$, respectively. Further, we assume a standard Gaussian $p(\boldsymbol{z_\epsilon}) = N(\boldsymbol{0}, \text{diag}(\boldsymbol{I}))$ to model *process noise* affecting the dynamical system $f_{\boldsymbol{\theta}}(\cdot)$, thus modeling approximations and integration errors. Though we assume a Gaussian distribution for convenience, more sophisticated alternative mechanisms for representing $\boldsymbol{z}$ can be considered, such as normalizing flows (Rezende and Mohamed, 2015).

**Black-box Dynamics** ODESolve is a solver that simulates the *state-time* matrix $X \in \mathbb{R}^{D \times T}$ (4) as the solution to the dynamics (3) at desired time points $\{t_0, \ldots, t_T\}$ given the initial state $\boldsymbol{x_0}$. We control the tradeoff between the accuracy of the simulated $X$ and the computational cost with a tolerance hyperparameter. Note that $X$ can be solved at arbitrary time-points, including irregularly sampled observations (see De Brouwer et al. (2019); Rubanova et al. (2019) for details). We specify the dynamics $f_{\boldsymbol{\theta}}(\cdot)$ using a multi-layer perceptron (MLP) and, following Chen et al. (2018),

we learn the parameters of $f_{\boldsymbol{\theta}}(\cdot)$ using the *adjoint sensitivity method*. Note that the recently proposed *stochastic* adjoint sensitivity method (Li et al., 2020) can be also considered for computational efficiency.

**Emission Process** In practice, observations $Y$ can be either non-negative or have a skewed distribution across a diverse range of applications such as those with biological signals, *e.g.*, heart-rate, temperature, blood pressure, *etc.* While non-skewed distributions such as the standard Gaussian are convenient, they are inappropriate for such observations, since they are typically characterized by a symmetric variance. So motivated, we wish to estimate a flexible (skewed) distribution by synthesizing observations $Y \sim p(Y|m_{\boldsymbol{\gamma}}(X), \sigma, \tau)$ from an asymmetric Laplace distribution (ALD) (Geraci and Bottai, 2007), where $0 < \tau < 1$, $\sigma > 0$, $-\infty < m_{\boldsymbol{\gamma}}(X) < \infty$, are skew, scale, and location parameters, respectively. The ALD is formulated as:

$$p_Y(Y; m_{\boldsymbol{\gamma}}(X), \sigma, \tau) = \frac{\tau(1 - \tau)}{\sigma} \times \quad (7)$$

$$\exp\left(-\left(\frac{Y - m_{\boldsymbol{\gamma}}(X)}{\sigma}\right)\left[\tau - I(Y \le m_{\boldsymbol{\gamma}}(X))\right]\right),$$

where $I(\cdot)$ is the indicator function. Note $m_{\boldsymbol{\gamma}}(\cdot)$ is a transformation that maps the state-time matrix $X$ to observations $Y$, s.t., $P(Y < m_{\boldsymbol{\gamma}}(X)) = \tau$, where $m_{\boldsymbol{\gamma}}(X)$ is the $\tau$-th quantile of the distribution. Consequently, learning $\{m_{\boldsymbol{\gamma}}(X)\}_{s=1}^S$ that corresponds to a set of $S$ quantiles $\{\tau\}_{s=1}^S$, provides a flexible approach for asymmetric uncertainty estimation. In our experiments we learn $\boldsymbol{\sigma}(t) \in \mathbb{R}^K$ and set $\tau = \{0.025, 0.50, 0.975\}$, so $S = 3$, thus effectively learning the median and 95% confidence intervals. However, alternatives such as the interquantile range, for which $\tau = \{0.25, 0.75\}$ are also possible.

## 2.2 LEARNING

We aim to maximize the joint marginal log-likelihood:

$$\max_{\boldsymbol{\theta}, \psi, \boldsymbol{\gamma}} \mathbb{E}_{Y, \boldsymbol{u} \sim \mathcal{D}} \log p_{\boldsymbol{\theta}, \psi, \boldsymbol{\gamma}}(Y, \boldsymbol{u}) =$$

$$\max_{\boldsymbol{\theta}, \psi, \boldsymbol{\gamma}} \mathbb{E}_{Y, \boldsymbol{u} \sim \mathcal{D}} \log \int p_{\boldsymbol{\theta}, \psi, \boldsymbol{\gamma}}(Y, \boldsymbol{u}, \boldsymbol{z}) d\boldsymbol{z}\,, \quad (8)$$

where we marginalize out the latent variable $\boldsymbol{z}$. For high-dimensional datasets and complex generative models such as neural networks, integration over the latent variables in (8) is intractable. Therefore, we introduce a variational posterior $q_{\boldsymbol{\varphi}, \boldsymbol{\phi}}(\boldsymbol{z}|Y, \boldsymbol{u})$ to approximate the true (but intractable) posterior $p(\boldsymbol{z}|Y, \boldsymbol{u})$ specified as a neural network with parameters $\{\boldsymbol{\varphi}, \boldsymbol{\phi}\}$.

**Posterior Distribution** Several variations for modeling $q_{\boldsymbol{\varphi}, \boldsymbol{\phi}}(\boldsymbol{z}|Y, \boldsymbol{u})$ consistent with assumed generative models have been proposed. For instance, Kingma et al. (2014);

Siddharth et al. (2017), assume a latent $\boldsymbol{u}$ and decomposition $q(\boldsymbol{z}, \boldsymbol{u}|Y) = q(\boldsymbol{z}|Y, \boldsymbol{u})q(\boldsymbol{u}|Y)$. However, such assumptions require *ad hoc* auxiliary objectives for efficiently learning from $\boldsymbol{u}$. Moreover, $q(\boldsymbol{z}|Y, \boldsymbol{u})$ does not capture relationships among the different input values or learn input-specific representations, which is crucial for mechanistic understanding and zero-shot learning. Fortunately, more recently, Joy et al. (2021) formulated a principled inference model that allows capturing input-specific representations by leveraging both Bayes' theorem and conditional independence $Y \perp\!\!\!\perp \boldsymbol{u}|\boldsymbol{z}$ (consistent with our assumed generative graph) via

$$q_{\boldsymbol{\varphi},\phi}(\boldsymbol{z}|Y, \boldsymbol{u}) = \frac{q_{\boldsymbol{\varphi}}(\boldsymbol{u}|\boldsymbol{z_u})q_\phi(\boldsymbol{z}|Y)}{q_{\boldsymbol{\varphi},\phi}(\boldsymbol{u}|Y)}, \qquad (9)$$

where $q_\phi(\boldsymbol{z}|Y)$ and $q_{\boldsymbol{\varphi}}(\boldsymbol{u}|\boldsymbol{z_u})$ are neural networks parameterized by $\{\boldsymbol{\varphi}, \phi\}$, and

$$q_{\boldsymbol{\varphi},\phi}(\boldsymbol{u}|Y) = \int q_{\boldsymbol{\varphi}}(\boldsymbol{u}|\boldsymbol{z_u})q_\phi(\boldsymbol{z}|Y)d\boldsymbol{z}. \qquad (10)$$

Moreover, we specify the variational distribution as Gaussian $q_\phi(\boldsymbol{z}|Y) = N\left(\boldsymbol{\mu}_{\boldsymbol{\psi}}(Y), \text{diag}\left(\boldsymbol{\sigma}_{\boldsymbol{\psi}}^2(Y)\right)\right)$ and categorical $q_{\boldsymbol{\varphi}}(\boldsymbol{u}|\boldsymbol{z_u}) = \text{Cat}\left(\boldsymbol{u}|\pi_{\boldsymbol{\varphi}}(\boldsymbol{z_u})\right)$, if $\boldsymbol{u}$ is discrete or Gaussian otherwise.

**Evidence Lower Bound**   Introducing (9) to approximate the posterior in (8) yields a tractable *evidence lower bound* (ELBO) for each observation as derived by Joy et al. (2021):

$$\log p_{\boldsymbol{\theta},\boldsymbol{\psi},\boldsymbol{\gamma}}(Y, \boldsymbol{u}) \geq \log q_{\boldsymbol{\varphi},\phi}(\boldsymbol{u}|Y) + \log p(\boldsymbol{u}) + \qquad (11)$$
$$\mathbb{E}_{q_\phi(\boldsymbol{z}|Y)}\left[\frac{q_{\boldsymbol{\varphi}}(\boldsymbol{u}|\boldsymbol{z_u})}{q_{\boldsymbol{\varphi},\phi}(\boldsymbol{u}|Y)}\log\left(\frac{p_{\boldsymbol{\theta},\boldsymbol{\psi},\boldsymbol{\gamma}}(Y|\boldsymbol{z})p_{\boldsymbol{\psi}}(\boldsymbol{z}|\boldsymbol{u})}{q_{\boldsymbol{\varphi}}(\boldsymbol{u}|\boldsymbol{z_u})q_\phi(\boldsymbol{z}|Y)}\right)\right],$$

where $\log p(\boldsymbol{u})$ is a constant, $\log q_{\boldsymbol{\varphi},\phi}(\boldsymbol{u}|Y)$ is a classification or regression conditional distribution formulation for $\boldsymbol{u}$ discrete or continuous, respectively, and $\frac{q_{\boldsymbol{\varphi}}(\boldsymbol{u}|\boldsymbol{z_u})}{q_{\boldsymbol{\varphi},\phi}(\boldsymbol{u}|Y)}$ are weights for the log-likelihood ratio we seek to maximize. We leverage the simulated *state-time* matrix $X$ *trajectories* from the ODESolver, as a means of constraining the mapping $\boldsymbol{z} \to Y$ in $p_{\boldsymbol{\theta},\boldsymbol{\psi},\boldsymbol{\gamma}}(Y|\boldsymbol{z})$ with learned dynamics $f_{\boldsymbol{\theta}}(\cdot)$ according to the emission process in (5) formulated as an ALD distribution in (7). We learn neural network parameters $\{\boldsymbol{\theta}, \boldsymbol{\psi}, \boldsymbol{\gamma}, \boldsymbol{\varphi}, \phi\}$ by maximizing the evidence lower bound (ELBO) in (11) via stochastic gradient descent.

**Theoretical Connections**   Assuming a perfectly disentangled latent space (Higgins et al., 2018), we propose a generative process that synthesizes observations $Y$ given system-inputs $\boldsymbol{u}$, subject to latent variable $\boldsymbol{z} = \{\boldsymbol{z_u}, \boldsymbol{z_\epsilon}\}$, which is a concatenation of independent sources of variation, *i.e.*, *input-specific* $\boldsymbol{z_u}$ and *noise-specific* $\boldsymbol{z_\epsilon}$. However, inferring the independent factors from posterior $q_{\boldsymbol{\varphi},\phi}(\boldsymbol{z}|Y, \boldsymbol{u})$ (9) without supervision is impossible in arbitrary generative

models (Locatello et al., 2019). Hence we leverage the formulation from (Joy et al., 2021), which naturally enables system-input inference $q_{\boldsymbol{\varphi},\phi}(\boldsymbol{u}|Y)$ consistent with our assumed data-generation model (see Figure 1), and without requiring additional *ad hoc* loss terms.

### 2.3   INFERENCE

ODE models are commonly used for observational data imputation, *i.e.*, interpolating or extrapolating tasks (Rubanova et al., 2019; Chen et al., 2018). For interpolation, ODE models generate an observation conditioned on values from a subset of time points $T_I \subseteq \{t_0, ..., t_T\}$ within the full-time interval $t \in [0, T]$. Moreover, for extrapolation tasks, the ODE model generates observations at future time points $t > T$, conditioned on values from previous times $t \in [0, T]$. Unlike previous works, here we focus on deeper understanding of system input effects, namely, $(i)$ synthesizing observations given latent variable sample $\boldsymbol{z}$ from the prior distribution in (2), and $(ii)$ inferring system inputs $\boldsymbol{u}$ given observations. Further, we consider the challenging *zero-shot learning* setup for synthesizing data from novel combinations or perturbations of system inputs.

## 3   RELATED WORK

**Variational Learning**   Recent machine learning research in variational inference for latent state-space models has benefited from advances in computational efficiency of integrating mechanistic models with observational data (Zenker et al., 2007). For instance, recently proposed neural ODEs (Rubanova et al., 2019) have enabled learning of continuous-time dynamics $f(\cdot)$ at low computational costs. For these latent state-space models, the estimation of model parameters is specified as a maximum-likelihood problem, where the dynamics are set as a constraint (González et al., 2014). Most approaches rely on amortized inference (Kingma and Welling, 2013; Rezende et al., 2014) to learn an intractable posterior (Linial et al., 2021; Roeder et al., 2019; Rubanova et al., 2019). However, these variational learning methods diverge in two main aspects: $i)$ proposed probabilistic graphical model, and $ii)$ assumptions needed to estimate $\{f(\cdot), m(\cdot)\}$, the dynamics and emission functions, respectively. Unlike existing approaches that assume a Gaussian emission process, the proposed method SL-ODE formulates a flexible quantile regression approach for capturing uncertainties in observational data. See Table 1 for an overview of the various modeling assumptions.

**Structured Latent-Space Representations**   Structured latent space modeling for nonlinear dynamical systems has been considered in the context of Kalman variational auto-encoders that retain the Markovian structure of hidden Markov models (Krishnan et al., 2017; Fraccaro et al.,

Table 1: Summary of related work. We categorize methods in terms of $(i)$ assumptions required for estimating $\{f(\cdot), m(\cdot)\}$, the ODE and emission functions, respectively, and $(ii)$ ability to perform tasks essential for the mechanistic understanding of system input effects: inferring of system inputs $\boldsymbol{u}$ given observations $Y$ and controlled generation of $Y$ given $\boldsymbol{u}$.

| Method | ODE function $f(\cdot)$ | Emission function $m(\cdot)$ | Predicts $\boldsymbol{u}$ | Controlled generation given $\boldsymbol{u}$ | Continuous-time | Asymmetric Uncertainty |
|---|---|---|---|---|---|---|
| UKF (Wan et al., 2001) | required | required | ✗ | ✗ | ✗ | ✗ |
| GOKU-net (Linial et al., 2021) | required | learned | ✓ | ✗ | ✓ | ✗ |
| Hierarchical-ODE (Roeder et al., 2019) | learned | required | ✗ | ✓ | ✓ | ✗ |
| DMM (Krishnan et al., 2017) | learned | learned | ✗ | ✗ | ✗ | ✗ |
| Latent-ODE (Rubanova et al., 2019) | learned | learned | ✗ | ✗ | ✓ | ✗ |
| SL-ODE (proposed) | learned | learned | ✓ | ✓ | ✓ | ✓ |

2017; Miladinović et al., 2019; Yingzhen and Mandt, 2018). Such latent state-space models focus on separating *static from dynamic* (Fraccaro et al., 2017; Yingzhen and Mandt, 2018), *domain-invariant from domain-specific* (Miladinović et al., 2019), and *position from momentum* (Yildiz et al., 2019) latent variables. Complementary to these methods, we do not impose the Markovian structure but instead propose to learn a principled structured variational posterior $q_{\boldsymbol{\varphi}, \boldsymbol{\phi}}(\boldsymbol{z}|Y, \boldsymbol{u})$ conditional on both observations $Y$ and system inputs $\boldsymbol{u}$, which we decompose according to (9). Our structured latent-space enables previously overlooked tasks essential for the mechanistic understanding of system input effects on dynamical systems: $(i)$ *controlled* generation of observations given system inputs, and $(ii)$ *inference* of system inputs from observations. Variational inference methods rarely account for *system inputs* except for Roeder et al. (2019); Linial et al. (2021). While Roeder et al. (2019) enables controlled generation, their formulation does not facilitate system input inference given observations, and though Linial et al. (2021) enables system input inference, controlled generation is not considered.

# 4 EXPERIMENTS

Below we provide details on the baseline methods considered for comparisons, the datasets employed, and the metrics used to evaluate our proposed approach. PyTorch code to replicate all experiments can be found at https://github.com/paidamoyo/structured_latent_ODEs. We summarize the SL-ODE training procedure, which is shared across all baseline methods except for the optimized evidence lower bound in Algorithm 1. See the SM for comprehensive details of the neural architectures of the baselines and proposed model.

## 4.1 BASELINES

For fair comparisons, i.e., all models use the same neural network architecture to model the ODE $f(\cdot)$, emission $m(\cdot)$, and encoder (maps observations $\boldsymbol{y}(t)$ to latent $\boldsymbol{z}$) functions. However, we preserve the assumed data generative process for each baseline. Recent state-of-the-art generative models for disentangled representations, *i.e.*, identifying independent factors of variation in data $Y$, leverage amortized infer-

ence (Locatello et al., 2019; Kim and Mnih, 2018). Therefore, we compare to competitive variational ODE-based baselines. We consider the following baselines:

- Latent-ODE: Gaussian latent variable model (Rubanova et al., 2019).
- GOKU-Net: Gaussian latent variable model accounting for system input inference (Linial et al., 2021).
- Hierarchical-ODE: Hierarchical latent variable model with conditional prior for system inputs (Roeder et al., 2019).

See Table 1 for a summary of the modeling assumptions in the baseline methods. Note that all baseline methods consider a Gaussian emission process, where the *observation noise* $\boldsymbol{\epsilon}(t)$ is shared across all observations. In contrast, our work adopts a flexible quantile regression approach formulated as an asymmetric Laplace distribution (7).

## 4.2 DATASETS

We perform evaluation on three biological datasets described below: $(i)$ CARDIOVASCULAR SYSTEM, $(ii)$ SYNTHETIC BIOLOGY, and $(iii)$ HUMAN VIRAL CHALLENGE.

**Human Viral Challenge** A real-world physiological dataset collected over multiple days from subjects equipped with Empatica E4 wearable wristband devices. On the second day, subjects were inoculated (challenged) with an H3N2 influenza pathogen, causing some to become infected, as clinically determined by viral shedding between 24 and 48 hours after inoculation. Moreover, peak symptoms usually occur, in average, 72 hours after inoculation. See She et al. (2020) for additional experimental details. We learn from 35 subjects' noisy time-series observations from four sensors $\boldsymbol{y}(t) = [\text{HR}, \text{TEMP}, \text{EDA}, \text{ACC}]$: heart rate (HR), temperature (TEMP), electrodermal activity (EDA), and accelerometer (ACC). Automated infection detection (*e.g.*, viral shedding) from a healthy baseline, around inoculation time and before shedding, has the potential to improve health awareness and is crucial in implementing effective infection prevention strategies. Hence, we evaluate our model on 5-fold cross-validation (due to small sample size) for subject outcome $\boldsymbol{u} = [u_1, u_2]$, where $u_1 \in \{0, 1\}$ and $u_2 \in \{0, 1\}$ indicates symptoms and viral shedding respectively.

Table 2: Performance comparisons for HUMAN VIRAL CHALLENGE via 5-fold cross-validation. System inputs $\boldsymbol{u}$ are binary outcomes indicating viral shedding and symptoms. We report methods without system input inference or controlled prior generation mechanisms as NA (not available).

| Method | $\boldsymbol{u}$ Accuracy (%) ↑ | $L_1$ error (posterior, prior) ↓ | ELBO ↑ |
|---|---|---|---|
| Latent-ODE | NA | (108.08, NA) | -362.48 |
| GOKU-Net | 0.66 | (91.97, NA) | -477.87 |
| Hierarchical-ODE | NA | (260.78, 347.97) | -426.43 |
| SL-ODE-Gaussian (ablation) | 0.63 | (88.86, 110.71) | -355.89 |
| SL-ODE (proposed) | **0.67** | (**39.73**, **40.3**) | **-327.73** |

**Cardiovascular System**  In a clinical setting, identification of system inputs $\boldsymbol{u}$ and states $\boldsymbol{x}(t)$ given noisy patient-specific clinical observations $\boldsymbol{y}(t)$ has the potential to improve differential diagnosis and predict responses to therapeutic interventions. As a result, several models for the cardiovascular system have been adapted in critical care environments, including a simplified cardiovascular system ODE model (Zenker et al., 2007), also recently considered in Linial et al. (2021). Following Linial et al. (2021) we generate ODE states $\boldsymbol{x}(t) = (SV(t), P_a(t), P_v(t), S(t))$ representing cardiac stroke volume (amount of blood ejected by the heart), arterial blood pressure, venous blood pressure, and autonomic baroreflex tone (reflex responsible for adapting perturbations in blood pressure and keeping homeostasis), respectively. We observe noisy sequences $\boldsymbol{y}(t) = (P_a(t), P_v(t), f_{\text{HR}}(t)) + \boldsymbol{\epsilon}(t)$, where $f_{\text{HR}}(t)$ is the patients heart-rate, and $\boldsymbol{\epsilon}(t)$ is the *observation* noise.

We wish to infer system inputs $\boldsymbol{u} = (I_{\text{external}}, R_{\text{TPR}_{\text{Mod}}})$ from 1000 time-series observations $\boldsymbol{y}(t)$, where $I_{\text{external}} < 0$ implies a patient is loosing blood, while $R_{\text{TPR}_{\text{Mod}}} < 0$ implies septic shock (*i.e.*, total peripheral resistance is getting low), resulting in four interpretable conditions:

- Healthy (both non-negative).
- Hemorrhagic shock ($I_{\text{external}} < 0, R_{\text{TPR}_{\text{Mod}}} \geq 0$).
- Distributive shock ($I_{\text{external}} \geq 0, R_{\text{TPR}_{\text{Mod}}} < 0$).
- Combined shock ($I_{\text{external}} < 0, R_{\text{TPR}_{\text{Mod}}} < 0$).

**Synthetic Biology**  The SYNTHETIC BIOLOGY case study is derived from a laboratory experimental dataset. Measurements are collected to model the dynamic behavior of genetically engineered devices in bacterial cell cultures with different combinations of shared genetic components. Characterization of cell culture response in genetic components given *experimental conditions* (or *treatments*) to generate desired responses for diagnostic, therapeutic, biotechnology applications, *etc.*, is time-intensive and unreliable (Nielsen et al., 2016). Therefore, we wish to learn a structured latent representation of the system inputs and observations to characterize novel devices consisting of combinations from select genetic components, *i.e.*, zero-shot learning, across different treatments. Below we summarize the dataset; see Roeder et al. (2019) for a detailed description including

---

**Algorithm 1** SL-ODE: Structured Latent ODE Model.

**Input**: ODE solver, Hyper-parameters
**Parameter**: Initialize parameters $\{\boldsymbol{\theta}, \psi, \boldsymbol{\gamma}, \boldsymbol{\varphi}, \phi\}$
**Output**: Maximize ELBO

1:  $\boldsymbol{z} \sim q_\phi(\boldsymbol{z}|Y)$ specified as Encoder $(\boldsymbol{y}(t); \phi)$
2:  $\boldsymbol{x}_0 = \text{InitState}(\boldsymbol{z}; \boldsymbol{\theta})$
3:  Simulate

$$X = \text{ODESolve}\left(f_{\boldsymbol{\theta}}, \boldsymbol{x}_0, (t_0, t_1, .., t_T)\right)$$

$$\text{s.t. } \frac{d\boldsymbol{x}}{dt} = f_{\boldsymbol{\theta}}(\boldsymbol{x}; \boldsymbol{z}, t).$$

4:  Reconstruct $Y \sim p\left(Y | m_{\boldsymbol{\gamma}}(X), \sigma, \tau\right)$
5:  Comptute ELBO

$$\mathbb{E}_{q_\phi(\boldsymbol{z}|Y)}\left[\frac{q_{\boldsymbol{\varphi}}(\boldsymbol{u}|\boldsymbol{z_u})}{q_{\boldsymbol{\varphi}, \phi}(\boldsymbol{u}|Y)} \log\left(\frac{p_{\boldsymbol{\theta}, \psi, \boldsymbol{\gamma}}(Y|\boldsymbol{z}) p_\psi(\boldsymbol{z}|\boldsymbol{u})}{q_{\boldsymbol{\varphi}}(\boldsymbol{u}|\boldsymbol{z_u}) q_\phi(\boldsymbol{z}|Y)}\right)\right]$$
$$ + \log q_{\boldsymbol{\varphi}, \phi}(\boldsymbol{u}|Y) + \log p(\boldsymbol{u})$$

6:  Backpropagate and update $\{\boldsymbol{\theta}, \psi, \boldsymbol{\gamma}, \boldsymbol{\varphi}, \phi\}$
7:  **return** solution

---

ODE dynamics. The system inputs $\boldsymbol{u} = [\boldsymbol{c}, \boldsymbol{g}]$, consist of two variables:

- A multi-hot vector representing different combinations of genetics components making up six genetic devices $\boldsymbol{g} \in \{Pcat\text{-}Pcat, RS100\text{-}S32, RS100\text{-}S34, R33\text{-}S32, R33\text{-}S175, R33\text{-}S34\}$.
- Different concentrations of chemicals (or treatments) $\boldsymbol{c} = \{C_6, C_{12}\}$.

Given the system inputs, we observe 312 noisy time-series observations captured from four optical devices $\boldsymbol{y}(t) = [\text{OD}, \text{RFP}, \text{YFP}, \text{CFP}]$: optical density (OD), red fluorescent protein (RFP), yellow fluorescent protein (YFP), and cyan fluorescent protein (CFP). We evaluate our model on two tasks: $(i)$ 4-fold cross-validation (due to small sample size) for *multiple device inference*, and $(ii)$ held-out (novel) device inference (*i.e., zero-shot learning*), which we evaluate on observations from $\boldsymbol{g} = R33\text{-}S34$ and $\boldsymbol{g} = R33\text{-}S32$.

Table 3: Performance comparisons for SYNTHETIC BIOLOGY data via 4-fold cross-validation *multiple device* inference task. System inputs $\boldsymbol{u} = [\boldsymbol{g}, \boldsymbol{c}]$, where $\boldsymbol{g}$ are categorical device genetic components and $\boldsymbol{c}$ are continuous treatment values. We report methods without system input inference or controlled prior generation mechanisms as NA.

| Method | $\boldsymbol{g}$ Accuracy (%) ↑ | $\boldsymbol{c}$ MSE ↓ | $L_1$ error (post, prior) ↓ | ELBO ↑ |
|---|---|---|---|---|
| Latent-ODE | NA | NA | (17.47, NA) | 880.83 |
| GOKU-Net | 90.71 | 1.34 | (5.08, NA) | 1411.61 |
| Hierarchical-ODE | NA | NA | (18.25, 18.17) | 896.07 |
| SL-ODE-Gaussian (ablation) | 91.07 | **0.87** | (5.58, 14.21) | 1296.11 |
| SL-ODE (proposed) | **92.95** | 0.98 | (**4.95**, **12.87**) | **1830.89** |

## 4.3 QUANTITATIVE ANALYSIS

Experimental results in Tables 2, 3, and 5 (in SM) demonstrate that the proposed SL-ODE consistently outperforms baseline methods across all evaluation metrics and datasets. We evaluate SL-ODE and compare to baseline methods on the following metrics:

- System input inference $\boldsymbol{u}$ given observational data $\boldsymbol{y}(t)$. We report accuracy and mean squared error (MSE) for categorical and continuous system inputs, respectively.
- We compare averaged system input-specific $L_1$ error from posterior or prior predictive distributions against ground truth observations. For the prior distribution, we evaluate methods capable of *controlled* generation given system inputs $\boldsymbol{u}$.
- Estimated evidence lower bound for model fit evaluation.

**Evidence Lower Bound (ELBO)** As expected, the latent-ODE model has the worst ELBO, since it is the only model that does not account for system inputs when modeling the posterior or prior distributions. Therefore, the model capacity is limited to a simple Gaussian posterior distribution. In contrast, our structured modeling approach has significant benefits over baseline methods in model fit (or ELBO), due to its system input inference (9) and structured conditional prior (2). Though Hierarchical-ODE assumes a conditional prior, it does not consider a system input inference mechanism. Moreover, while GOKU-Net considers a system input inference mechanism, it is constrained by its Gaussian prior.

**Posterior and prior predictive distributions $L_1$ error** Formulated as an absolute difference between *input-specific* predictions and ground truth averaged across observations and system inputs. Our model achieves the lowest posterior and prior distributions $L_1$ error across all datasets. However, we noticed a drop in performance between the SYNTHETIC BIOLOGY posterior and prior errors. We attribute the performance decline to the challenge associated with accounting for complex system inputs, *i.e.*, heterogeneous (mixture of categorical and continuous) variables. Note that we do not report the prior $L1$ error on GOKU-Net and Latent-ODE since these models do not consider controlled generation given system inputs.

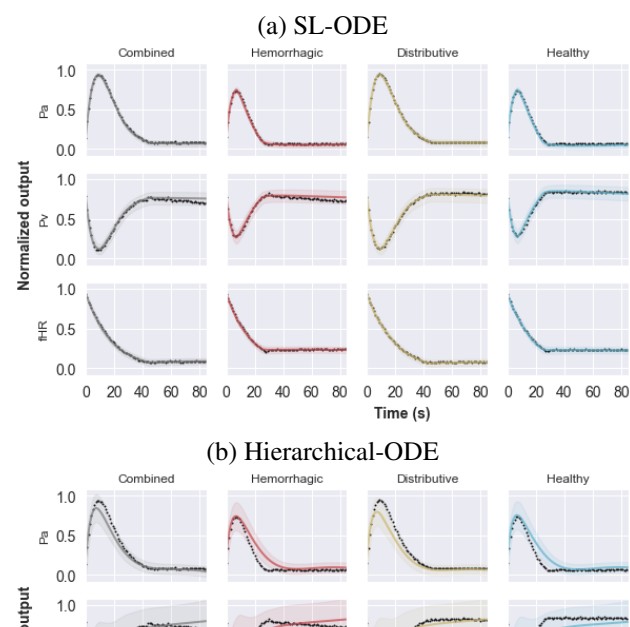

Figure 2: Ground truth (black) *vs. controlled* generated observations (colored) given system inputs $\boldsymbol{u}$ according to assumed prior for (a) proposed SL-ODE and (b) Hierarchical-ODE models on CARDIOVASCULAR SYSTEM data. We average synthesized observational data $\boldsymbol{y}(t)$ across all class-specific time series and report the estimated median with 95% confidence interval (CI).

**System input inference** We report a competitive advantage over GOKU-Net in SYNTHETIC BIOLOGY and HUMAN VIRAL CHALLENGE system input inference, owing to our structured conditional prior representations (2), which is not considered in GOKU-Net. Note that we do not report results on Hierarchical-ODE and Latent-ODE methods, which do not consider system input inference.

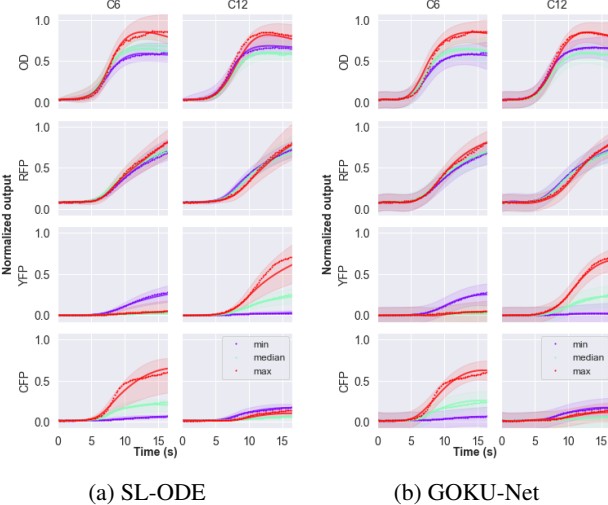

(a) SL-ODE      (b) GOKU-Net

Figure 3: Posterior predictive distribution on SYNTHETIC BIOLOGY data via 4-fold cross-validation *multiple device* inference task for (a) proposed SL-ODE and (b) GOKU-Net models. For clarity, we plot ground truth (dotted) time-series against median predictions (solid) across three $c = [C_6, C_{12}]$ treatments (minimum, median, and maximum), *e.g.*, when $C_6$= minimum, output is averaged across all $C_{12}$. Shaded areas indicate the predicted 95% CI.

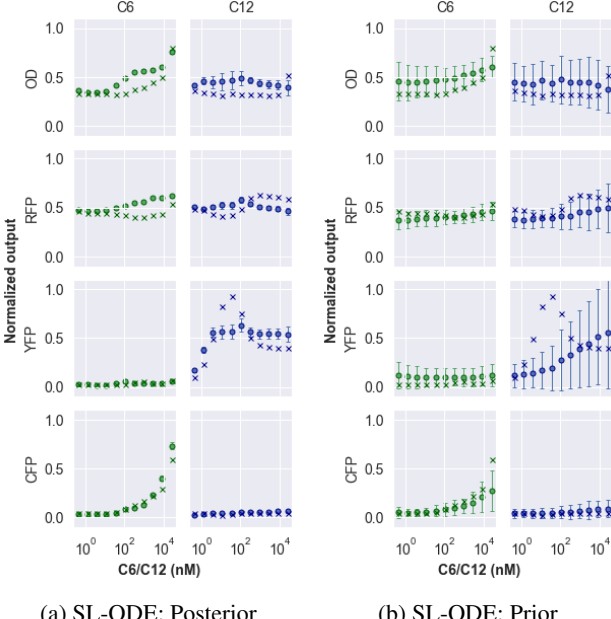

(a) SL-ODE: Posterior      (b) SL-ODE: Prior

Figure 4: SL-ODE SYNTHETIC BIOLOGY *held-out device* ($g = R33\text{-}S34$) task. Ground truth *vs.* (a) posterior predictive distribution and (b) *controlled* generated observations given system inputs $u = [g, c]$ according to assumed prior distribution (2). We plot the median (circles) with 95% CI against ground truth observations (crosses) averaged (200 $z$ samples) across all observations at the final time-point sweeping all $c = [C_6, C_{12}]$ treatments.

## 4.4 QUALITATIVE ANALYSIS

We further compare against the best performing baseline methods Hierarchical-ODE and GOKU-Net in Figures 2 and 3, respectively. Figure 2 demonstrates that the controlled generated samples from the assumed prior distribution of SL-ODE match the ground truth class-specific time-series better than samples from Hierarchical-ODE on the CAR-DIOVASCULAR SYSTEM dataset. Moreover, the estimated 95% CI of SL-ODE exhibit low-variance predictions. Similarly, in Figure 3 we present low-variance predictions at earlier times than GOKU-Net on the SYNTHETIC BIOLOGY dataset *multi-device* task per reported 95% CI posterior predictive distributions. See Figure 5 (in SM) for complete *multiple device* inference results across all methods. We observe a similar trend on the HUMAN VIRAL CHALLENGE dataset (see Figures 7-10 in SM), albeit capturing imperfect dynamics limited by the ODE class. This demonstrates that our quantile regression emission formulation (7) has a competitive advantage for capturing flexible and asymmetric uncertainties over the typical choice of standard Gaussian emission process. Additionally, the ablation study illustrates that the proposed SL-ODE with an *asymmetric* Laplace likelihood (7) has a quantitative competitive advantage over the alternative (SL-ODE-Gaussian) with Gaussian likelihood.

Finally, Figure 4 shows posterior and prior predictive summaries on the challenging SYNTHETIC BIOLOGY *held-out device* task (so-called zero-shot learning) across all treat-

ment values. Interestingly, except for mid $C_{12}$ treatments from YFP, SL-ODE closely matches ground truth observations for the posterior and prior predictive distributions. Accurately synthesizing data under novel input combinations is crucial for experimental science, where obtaining data is typically expensive and time consuming. We anticipate performance gains with additional training data from an S34 device component known to bind to $C_{12}$ (Roeder et al., 2019). See Figure 6 (in SM) for additional zero-shot learning results from held-out device $g = R33\text{-}S32$.

## 5 CONCLUSIONS

We have presented a principled statistical framework for integrating mechanistic models with amortized inference. We applied this framework to a constrained maximum-likelihood estimation of time-series observational data and static system inputs. Moreover, we demonstrated the benefits of capturing *system input-specific* variations in the latent space for a deeper understanding of system input effects on dynamical systems. Further, the proposed inference method does not assume known ODE dynamics or emission functions. Unlike prior works that presume a Gaussian emission process, we quantify *observation noise* with quantile re-

gression for flexible (skewed) uncertainty estimation. We presented results on three challenging biological datasets, characterizing human physiological event states, cardiovascular systems, and genetically engineered devices in synthetic biology. We demonstrated significant performance gains over competitive baselines in uncertainty estimation and mechanistic understanding tasks: *controlled* generation of observational data given novel system input combinations, and *inference* of biologically meaningful inputs from observational data. In the future, we plan to extend our structured representation formulation to account for time-varying system inputs, frequently encountered in several dynamical systems, such as gene regulation (Calderhead et al., 2009). Finally, current research aims to account for irregularly sampled observations (De Brouwer et al., 2019; Rubanova et al., 2019), these approaches may also augment the scope of the proposed structured latent ODE model.

## Acknowledgements

The authors would like to thank the anonymous reviewers for their insightful comments. This research was supported by NIH/NINDS 1R61NS120246, NIH/NIDDK R01-DK123062, and ONR N00014-18-1-2871-P00002-3.

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

# A ADDITIONAL RESULTS

Figure 5 and Figures 7-10 provide all qualitative visualizations of the posterior predictive distributions across all methods on SYNTHETIC BIOLOGY and HUMAN VIRAL CHALLENGE datasets. Note that for fair comparisons, Hierarchical-ODE preserves the data generating graphical model of Roeder et al. (2019) but deviate in dynamics and emission functions, resulting in significantly worse performance than reported in Roeder et al. (2019). Additionally, we present results from held-out device posterior predictive distribution and controlled generated observations from novel device $g = R33\text{-}S32$ in Figure 6. See Table 5 for CARDIOVASCULAR SYSTEM quantitative results.

# B EXPERIMENTAL SETUP

Below we provide details of the neural-network architectures, selected hyper-parameters and pseudo-code for the proposed SL-ODE algorithm.

## B.1 NEURAL-NETWORK ARCHITECTURES

In all experiments, SL-ODE (proposed), GOKU-Net, Latent-ODE, and Hierarchical-ODE share the ODE $f(\cdot)$, emission $m(\cdot)$, and encoder (maps observations $\boldsymbol{y}(t)$ to latent $\boldsymbol{z}$) functions, detailed below. In general, we specify two-layer multi-layer perceptrons (MLPs) with 25 hidden units and Rectified Linear Unit (ReLU) as activation functions. Additionally, we implement 2-layer MLPs for the *system input-specific* distributions:

- Prior distribution $p_{\psi}(\boldsymbol{z_u}|\boldsymbol{u})$ used in SL-ODE and Hierarchical-ODE.
- Variational distribution $q_{\boldsymbol{\varphi}}(\boldsymbol{u}|\boldsymbol{z_u})$ used in SL-ODE and GOKU-Net.

**Encoder** Following Roeder et al. (2019), we apply a $1D$ CNN to observations $\boldsymbol{y}(t) \rightarrow$ average pooling $\rightarrow$ two-layer MLPs $\rightarrow$ latent variable $\boldsymbol{z}$ described with mean $\boldsymbol{\mu}$ and variance $\text{diag}(\boldsymbol{\sigma}^2)$. Note that the Hierarchical ODE model has an additional 2-layer MLP mapping system inputs to an input-specific latent variable.

**Black-box Dynamics** We leverage the *adjoint solver* Chen et al. (2018) to simulate the state-time matrix $X$ where the dynamics $f_{\boldsymbol{\theta}}(\cdot)$ are 2-layer MLPs with *Sigmoid* output-layer activations. Following Roeder et al. (2019), we specify dynamics as

$$\frac{d\boldsymbol{x}}{dt} = f_1(\boldsymbol{x}, \boldsymbol{z}, t; \theta) - \boldsymbol{x} \odot f_2(\boldsymbol{x}, \boldsymbol{z}, t; \theta),$$

where $\odot$ is the Hadamard product. Further, we initialize the initial state $\boldsymbol{x}_0$ as $\boldsymbol{z} \rightarrow$ 2-layer MLPs with *Sigmoid* output activation $\rightarrow \boldsymbol{x}_0$.

**Emission** We map the states $X$ to the observations $Y$ with a 1-layer linear MLP. For all baseline methods, the emission function outputs observation means $\boldsymbol{m}(t)$ and variances $\boldsymbol{\epsilon}(t)$. In contrast, our proposed approach (SL-ODE), outputs the median $\boldsymbol{m}(t)$, upper- $\boldsymbol{u}(t)$, and lower- $\boldsymbol{l}(t)$ quantiles according to the specified $\tau$.

## B.2 HYPER-PARAMETER SELECTION

We use the Adam optimizer (Kingma and Ba, 2015) with the following hyper-parameters: first moment 0.9, second moment 0.99, and epsilon $1 \times 10^{-8}$. We train all models using one NVIDIA P100 GPU with 16GB memory. See Table 4 for data-specific hyper-parameters. We split the CARDIOVASCULAR SYSTEM data into training, validation, and test sets as 80%, 10%, and 10% partitions, respectively. Further, we use the validation set for early stopping and learning model hyper-parameters. However, for the SYNTHETIC BIOLOGY and HUMAN VIRAL CHALLENGE datasets, we perform $k$-fold cross-validation due to the small sample sizes.

Table 4: Summary of data-specific hyper-parameters.

| Hyper-parameter | SYNTHETIC BIOLOGY | CARDIOVASCULAR SYSTEM | HUMAN VIRAL CHALLENGE |
|---|---|---|---|
| Mini-batch size | 36 | 128 | 28 |
| Learning rate | $3 \times 10^{-4}$ | $1 \times 10^{-3}$ | $1 \times 10^{-3}$ |
| States dimension ($D$) | 8 | 5 | 5 |

Table 5: Performance comparisons for CARDIOVASCULAR SYSTEM on test data. System inputs $u$ are interpretable patient states. We report methods without system input inference or controlled prior generation mechanisms as NA.

| Method | $u$ Accuracy (%) $\uparrow$ | $L_1$ error (posterior, prior) $\downarrow$ | ELBO $\uparrow$ |
|---|---|---|---|
| Latent-ODE | NA | (6.95, NA) | 9.12 |
| GOKU-Net | **100** | (5.06, NA) | 324.81 |
| Hierarchical-ODE | NA | (4.25, 4.42) | 374.94 |
| SL-ODE-Gaussian (ablation) | **100** | (0.66, 0.67) | 561.29 |
| SL-ODE (proposed) | **100** | (**0.56**, **0.57**) | **752.23** |

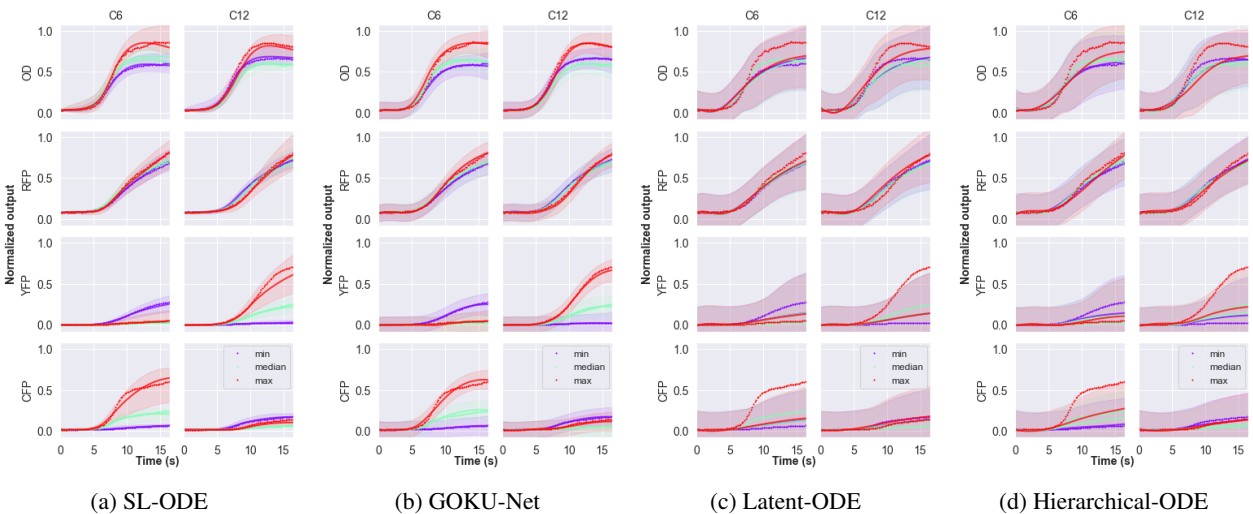

(a) SL-ODE      (b) GOKU-Net      (c) Latent-ODE      (d) Hierarchical-ODE

Figure 5: Posterior predictive distribution on SYNTHETIC BIOLOGY data via 4-fold cross-validation *multiple device* inference task for (a) proposed SL-ODE, (b) GOKU-Net, (c) Latent-ODE, and (d) Hierarchical-ODE models. For clarity, we plot ground truth (dotted) time-series against median predictions (solid) across three $c = [C_6, C_{12}]$ treatments (minimum, median, and maximum), *e.g.*, when $C_6$= minimum, output is averaged across all $C_{12}$. Shaded areas indicate the predicted 95% confidence interval (CI).

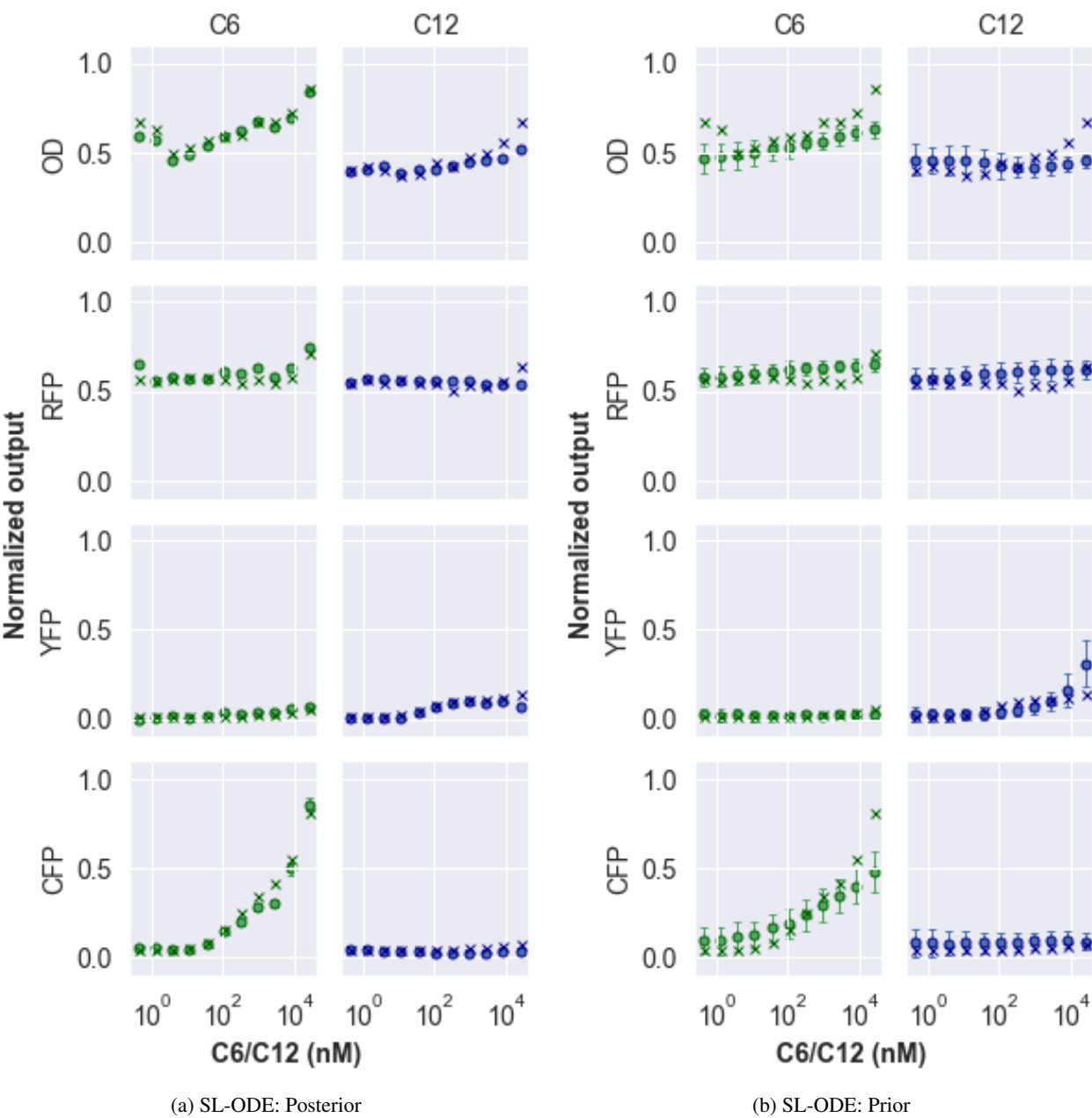

(a) SL-ODE: Posterior

(b) SL-ODE: Prior

Figure 6: SL-ODE SYNTHETIC BIOLOGY *held-out device* ($g = R33\text{-}S32$) task. Ground truth *vs.* (a) posterior predictive distribution and (b) *controlled* generated observations given system inputs $u = [g, c]$ according to assumed prior distribution. We plot the median (circles) with 95% CI against ground truth observations (crosses) averaged (200 $z$ samples) across all observations at the final time-point sweeping all $c = [C_6, C_{12}]$ treatments.

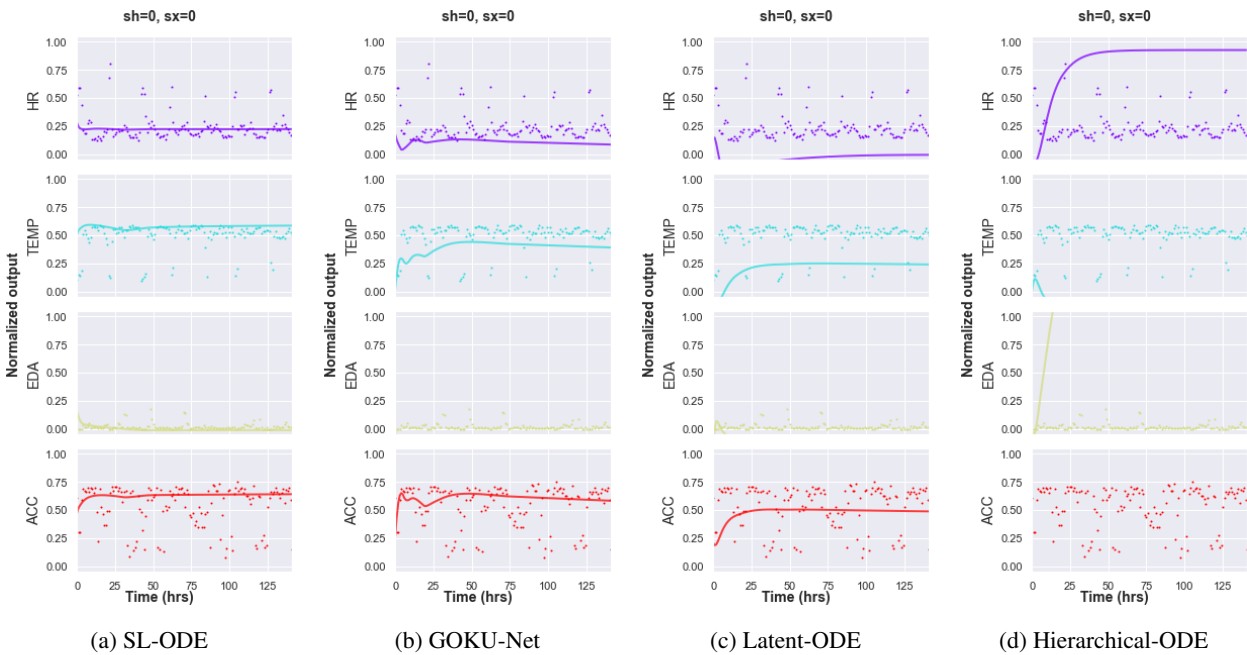

(a) SL-ODE  (b) GOKU-Net  (c) Latent-ODE  (d) Hierarchical-ODE

Figure 7: Posterior predictive distribution on HUMAN VIRAL CHALLENGE for randomly selected test patient showing one of the four combination binary outcomes $u$ for viral shedding (sh=0) and symptoms (sx=0) onset (a) proposed SL-ODE, (b) GOKU-Net, (c) Latent-ODE, and (d) Hierarchical-ODE models. For clarity, we plot ground truth (dotted) time-series against median predictions (solid). We do not show error bars since they are too large due to noisy data.

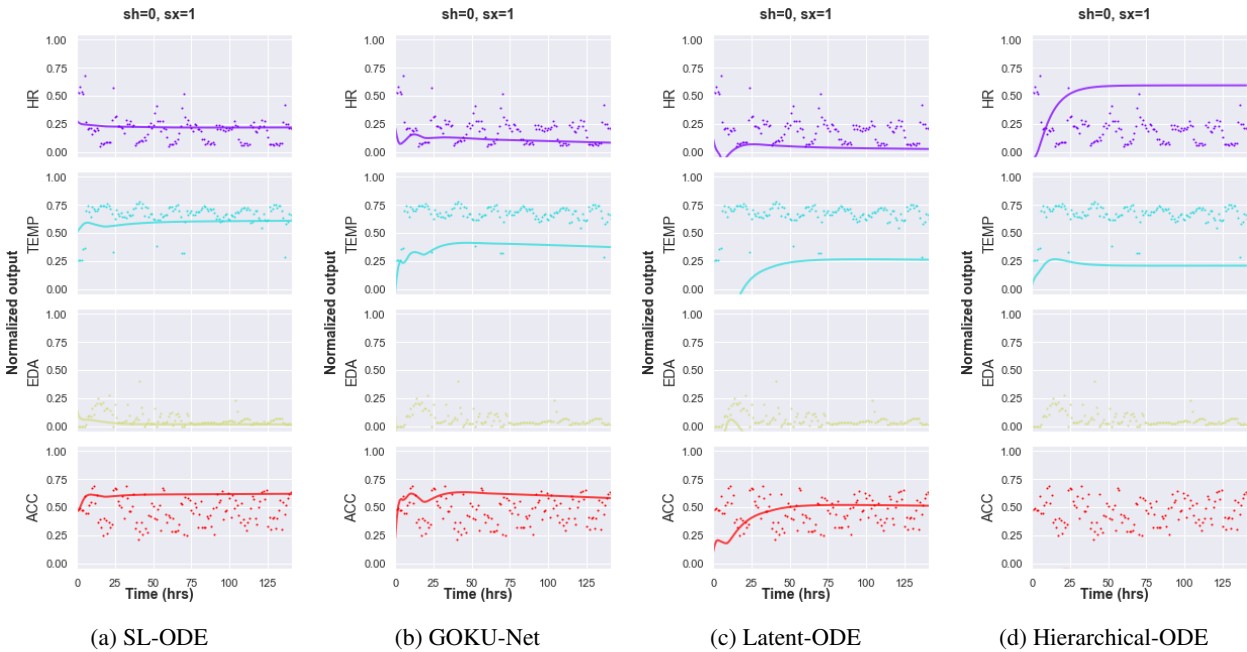

(a) SL-ODE  (b) GOKU-Net  (c) Latent-ODE  (d) Hierarchical-ODE

Figure 8: Posterior predictive distribution on HUMAN VIRAL CHALLENGE for randomly selected test patient showing one of the four combination binary outcomes $u$ for viral shedding (sh=0) and symptoms (sx=1) onset (a) proposed SL-ODE, (b) GOKU-Net, (c) Latent-ODE, and (d) Hierarchical-ODE models. For clarity, we plot ground truth (dotted) time-series against median predictions (solid). We do not show error bars since they are too large due to noisy data.

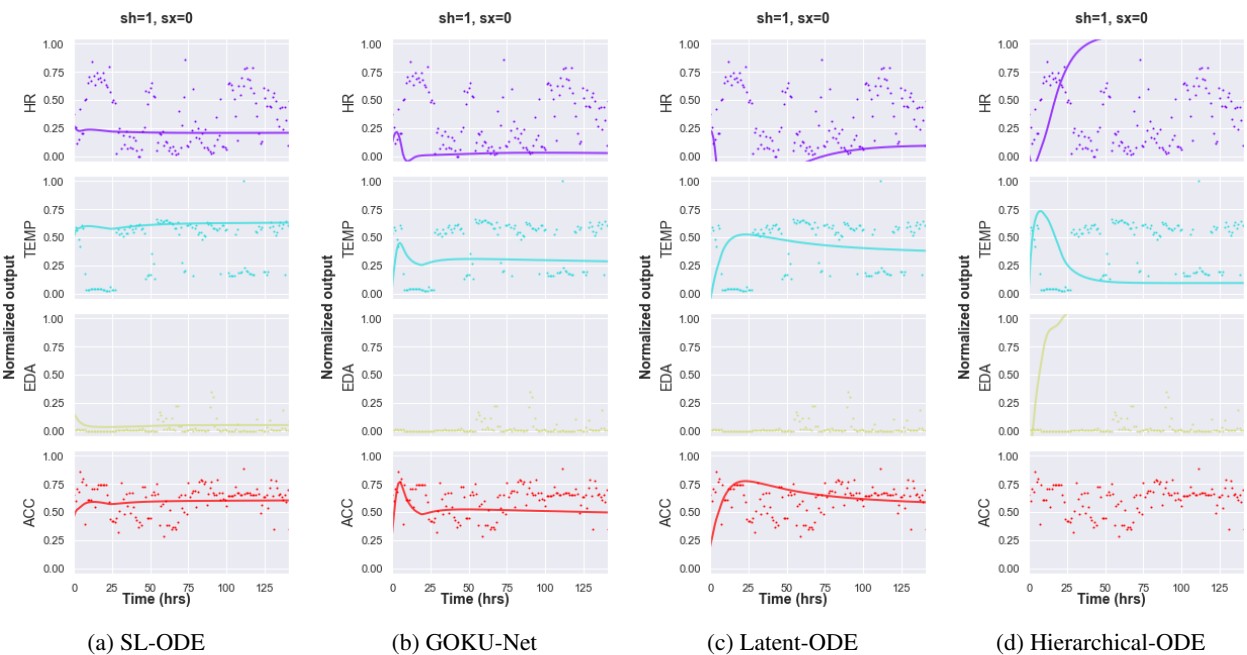

(a) SL-ODE      (b) GOKU-Net      (c) Latent-ODE      (d) Hierarchical-ODE

Figure 9: Posterior predictive distribution on HUMAN VIRAL CHALLENGE for randomly selected test patient showing one of the four combination binary outcomes $u$ for viral shedding (sh=1) and symptoms (sx=0) onset (a) proposed SL-ODE, (b) GOKU-Net, (c) Latent-ODE, and (d) Hierarchical-ODE models. For clarity, we plot ground truth (dotted) time-series against median predictions (solid). We do not show error bars since they are too large due to noisy data.

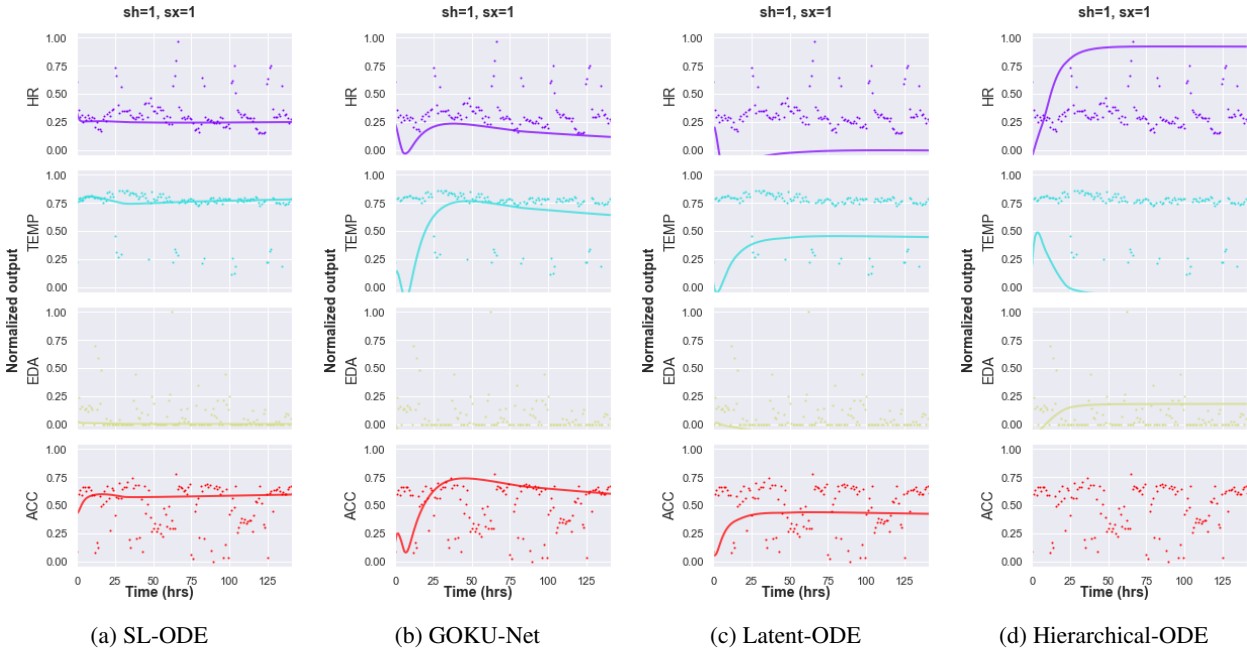

(a) SL-ODE      (b) GOKU-Net      (c) Latent-ODE      (d) Hierarchical-ODE

Figure 10: Posterior predictive distribution on HUMAN VIRAL CHALLENGE for randomly selected test patient showing one of the four combination binary outcomes $u$ for viral shedding (sh=1) and symptoms (sx=1) onset (a) proposed SL-ODE, (b) GOKU-Net, (c) Latent-ODE, and (d) Hierarchical-ODE models. For clarity, we plot ground truth (dotted) time-series against median predictions (solid). We do not show error bars since they are too large due to noisy data.