# OpenReview forum: "Capturing Actionable Dynamics with Structured Latent Ordinary Differential Equations"
_auai.org/UAI/2022/Conference — UAI 2022 Poster_

### Official Review · Reviewer_tfxk · 2022-04-09

**Q2(1) Originality/Novelty:** 3
**Q2(2) Significance/Impact:** 3
**Q2(3) Correctness/Technical Quality:** 3
**Q2(6) Clarity Of Writing:** 3
**Q6 Overall Score:** 6
**Q8 Confidence In Your Score:** 3

**Q1 Summary And Contributions:**

This paper proposed an method to learn the structured latent space representation of a dynamical system, the generative approach allows the model to connect system input, observation and latent space. The method is applied on three biological datasets and it outperforms prior work.



**Q2 Assessment Of The Paper:**

More detailed information regarding each of these aspects is given below:

**Q2(4) Quality Of Experiments (Optional):**

2: Fair: The experimental evaluation is weak: important baselines are missing, or the results do not adequately support the main claims.

**Q2(5) Reproducibility:**

2: Fair: Key resources (e.g., proofs, code, data) are unavailable but key details (e.g., proof sketches, experimental setup) are sufficiently well-described for an expert to confidently reproduce the main results.

**Q3 Main Strengths:**

1.The idea of generative learning and design of parameterization are novel.
2.The experiments illustrate the power of the proposed method



**Q4 Main Weakness:**

1.I’m not fully convinced by the latent variable setup $z={z_u,z_{\epsilon}}$, where the former is related to input u and latter is for noise. Does it mean the latent space does not depend on initial state $x_0$? If so, from Algorithm 1 how do we get $x_0$ from z?
2.This is not major concern since the experiments are thorough but all three are on biological field, it would be interesting to see how the model behave on other area. Even an toy example are useful.

**Q5 Detailed Comments To The Authors:**

Minor presentation issues:
The definition of $z_{u}$ and $z_{\epsilon}$ in the 2nd paragraph of section 2.1 can be moved to after equation (5), since both of them already appear in equation (2).

In line 5 of algorithm 1, the ELBO equation could be written in the sampling form(like \sum over examples).

Typo:
In line 5 of algorithm 1: Comptute->compute

**Q7 Justification For Your Score:**

I think overall the idea is novel and elegant. While my major concern is the setup of latent space. The presentation could be more detailed in algorithm implementation. The experiments are good but I'd like to see some toy examples to illustate the effectiveness of the model.

**Q9 Complying With Reviewing Instructions:**

1: Yes.

---

### Official Review · Reviewer_hnAG · 2022-04-12

**Q2(1) Originality/Novelty:** 1
**Q2(2) Significance/Impact:** 1
**Q2(3) Correctness/Technical Quality:** 3
**Q2(6) Clarity Of Writing:** 3
**Q6 Overall Score:** 4
**Q8 Confidence In Your Score:** 4

**Q1 Summary And Contributions:**

This papers proposes a VAE approach to modelling dynamical systems based on observed time series. Emphasize is put on being able to model systems controlled by an input $u$, and on interpretability. The architecture incorporates two types of latent variables, input specific and noise specific. The dynamics is simulated by and ODE solver parameterized by a neural network. The model is trained on several dataset and compared to several baselines.

**Q2 Assessment Of The Paper:**

More detailed information regarding each of these aspects is given below:

**Q2(4) Quality Of Experiments (Optional):**

2: Fair: The experimental evaluation is weak: important baselines are missing, or the results do not adequately support the main claims.

**Q2(5) Reproducibility:**

2: Fair: Key resources (e.g., proofs, code, data) are unavailable but key details (e.g., proof sketches, experimental setup) are sufficiently well-described for an expert to confidently reproduce the main results.

**Q3 Main Strengths:**

- The paper is well written,
- the model description is sound,
- the extensive use of real world data (biological dataset) is interesting.

**Q4 Main Weakness:**

- I tend to think the experiments do not support well the main claims state as "Understanding the effects of these system inputs on system outputs is crucial to have any meaningful model of a dynamical system" in the abstract and "Our
structured latent-space enables previously overlooked tasks essential for the mechanistic understanding of system input
effects on dynamical systems: (i) controlled generation of observations given system inputs, and (ii) prediction of system inputs from observations." Indeed, the validation is based on a purely predictive approach. There is, as far as I can see, no attempt to predict interventional data based on purely observational data, and no attempt to identify the ground truth causal structure of the system.
- the algorithmic approach is a combination of well known variational inference technique. While there are two levels of variational inference shown in Fig. 1, this does not appear to yield specific challenges.


**Q5 Detailed Comments To The Authors:**

I would suggest investigating in simulations whether the proposed approach is able to capture the latent structure of ground truth models. It has been demonstrated that VAEs do not provide, in general, identifiability guaranties, and as such it is unclear at this point in which case the proposed method provides a trustable latent structure.



**Q7 Justification For Your Score:**

Mostly the incremental aspect of the methodology and the inadequacy of the experiments to support main claims suggest that the paper either needs to reframe its objectives, or perform very different analyses to support the claims.

**Q9 Complying With Reviewing Instructions:**

1: Yes.

---

### Official Review · Reviewer_A6TN · 2022-04-15

**Q2(1) Originality/Novelty:** 2
**Q2(2) Significance/Impact:** 2
**Q2(3) Correctness/Technical Quality:** 3
**Q2(6) Clarity Of Writing:** 3
**Q6 Overall Score:** 6
**Q8 Confidence In Your Score:** 3

**Q1 Summary And Contributions:**

This paper proposes a framework for maximum likelihood learning of ODE-based models, building on similar suggestions from the literature. The main contribution is a more thorough treatment of static system inputs, by explicitly modelling its effect on latent state, enabling both generation of observations and inference of inputs. A second contribution concerns a more flexible model for emissions, which the authors motivate as particuarly suitable for certain applications.

**Q10 Ethical Concerns (Optional):**

No.

**Q2 Assessment Of The Paper:**

More detailed information regarding each of these aspects is given below:

**Q2(4) Quality Of Experiments (Optional):**

3: Good: The experimental evaluation is adequate, and the results convincingly support the main claims.

**Q2(5) Reproducibility:**

2: Fair: Key resources (e.g., proofs, code, data) are unavailable but key details (e.g., proof sketches, experimental setup) are sufficiently well-described for an expert to confidently reproduce the main results.

**Q3 Main Strengths:**

The motivation behind the paper is clearly made and makes sense: as the authors state, placing attention on system inputs has potential for significant impact on experimental design, as well as more useful learning outcomes. One clear benefit is that the proposed framework enables a set of tasks which none of the previous models could individually address completely.

The authors situate their work well within the literature and consider relevant tasks in the experimental evaluation, on three different and potentially challenging examples.

The technical content of the paper appears correct, and the paper is generally well-organised and written clearly, particularly the first half (introduction, model description, related work).

**Q4 Main Weakness:**

I am afraid that I was not entirely convinced that the potential impact promised is realised.

From the experimental results, the main gains appear to be in the predictive fit to outputs (and even there, the synthetic biology results in Figure 3 show very similar behaviour to the "competitor" GOKU-Net), while the inference of inputs is only marginally improved (with the exception of $c$ in the synthetic biology example).

I found parts of the Experiments section confusing (see detailed comments) and believe that they would benefit from further information, for the sake of both clarity and reproducibility.

My impression is that the contributions are not significantly novel, but note that I am not familiar with the previous work cited.

Finally, I disagree with the recurring claim that the framework provides a mechanistic, rather than statistical, understanding of the system, given that the mechanics are modelled as neural networks and not explicitly.

**Q5 Detailed Comments To The Authors:**

A main doubt I had is about the actual benefits of modelling the inputs separately, especially since they are static, rather than as latent states or the initial $\mathbf{x}_0$. Of the three examples, only in the synthetic biology one do the inputs feel like actual external system inputs. Perhaps this is due to conceptual confusion on my part. On a related issue, I believe it would be worth including some sort of analysis or at least acknowledgement of sensitivity to inputs or identifiability. For example, for the other methods that return worse estimates of inputs, do those estimates actually provide significantly different behaviour to the ground truth, or not?

As mentioned above, I had some difficulty understanding the experimental results and differentiating between the metrics compared. I believe that giving explicit formulas or more detailed descriptions, at least in the supplementary material, would be helpful for readers, not least those who want to reproduce the results. In Figures 2, 3 and 4, it is a bit confusing to have "Normalized output" next to input labels, which at first suggested to me that the plots were showing "predicted" inputs. Perhaps it would be clearer to swap the rows and columns? (unless I am misunderstanding something). In Figure 3, why are there 2 rows of plots for each method? My interpretation was that there are six treatment regimes depcited, such that only one of C_6 and C_12 is present in each (and in one of the three concentrations considered). Even if that is correct (and I am not sure it is?), it would be good to explain it, as it seemingly contradicts the caption. I admit that I am still confused as to what the two sets of plots show in Figure 4 - what data are the posterior predictive distributions conditioned on compared to the prior/controlled observations? (and the use of "prior" here may be confusing in itself)

On the topic of reproducibility, while it would be ideal to have included the code alongside the submission for review, your statement that you are willing to provide it publicly is very welcome. I hope that this will include not only the PyTorch code, but also the experimental evaluation. I am sure that this would be helpful both to those wanting to understand the details of the experiments, and to those performing the same or a similar analysis.

More minor comments:

Section 2.1:

"initial state initialization": phrasing seems confusing but I may be missing something

"Without loss of generality" (before Eq. 6): are there really no constraints introduced?

"thus capturing observational noise" (after Eq. 6): isn't this included in the emission process $m_\gamma$ instead? (the last paragraph on page 8 also seems to support that)

Section 2.2:

Eq. 8: repeat the $\max$ on the rhs

"However, such assumptions require ad hoc auxiliary objectives for efficiently learning from $u$": I do not understand this sentence.

"We learn network parameters ..." (end of "Evidence Lower Bound" paragraph): are $\phi$ and $\varphi$ also NN parameters?

Section 2.3:

"we focus on mechanistic understanding": As mentioned above, I do not agree with this phrasing; replacing a process with a NN gives a statistical understanding at best.

"predicting system inputs": Perhaps this is just me, but I find the phrasing counterintuitive and a bit confusing, since the inputs conceptually predate the state evolution and outputs of the system, and there is a notion of time in the models considered. I would suggest "inferring" or "synthesising", but acknowledge that this may just be me being weird.

Section 4.1:

"For fair comparisons ... ODE-based baselines": I am confused by this whole first paragraph: what does preserving vs sharing mean here? And I don't fully get the "Therefore" implication.

Table 1 caption: "We categorize methods in terms...": missing "of"

Table 2: it may be better to show the unnegated ELBO, for consistency with Table 3?

**Q7 Justification For Your Score:**

This is an interesting, if not groundbreaking, idea. The experimental evaluation left me with some doubts about its full potential, and I believe it could benefit from clarification. On the other hand, it appears technically sound, and does deliver improved performance in some regards. I have given my final score on the basis of correctness, general clarity and the potential for inspiring future work in this direction, as well as the fact that I am not well-equipped to judge the novelty.

**Q9 Complying With Reviewing Instructions:**

1: Yes.

---

### Official Review · Reviewer_W3Sp · 2022-04-17

**Q2(1) Originality/Novelty:** 3
**Q2(2) Significance/Impact:** 2
**Q2(3) Correctness/Technical Quality:** 3
**Q2(6) Clarity Of Writing:** 4
**Q6 Overall Score:** 7
**Q8 Confidence In Your Score:** 3

**Q1 Summary And Contributions:**

The paper proposes a generative structured latent ODE model that is more flexible compared to current models. Compared to Roeder et al.(2019) the proposed model allows for 1) A generalized latent structure over the model inputs (along with a noise term); 2) and uses an ALD as the emission process. A variational inference approach from Joy et al. is used for learning. Further, an uncertainty quantification framework and zero-shot learning setup are also considered.

**Q2 Assessment Of The Paper:**

More detailed information regarding each of these aspects is given below:

**Q2(4) Quality Of Experiments (Optional):**

4: Excellent: The experimental evaluation is comprehensive and the results are compelling.

**Q2(5) Reproducibility:**

4: Excellent: Key resources (e.g., proofs, code, data) are available and key details (e.g., proof sketches, experimental setup) are comprehensively described for competent researchers to confidently and easily reproduce the main results.

**Q3 Main Strengths:**

Apart from being more flexible, the proposed framework combines different features like 1) Input prediction 2) Data-generation given new inputs, 3) Continuous-time, etc. into a single unified framework while marginally outperforming the current methods in empirical evaluation. Further, the uncertainty quantification and the zero-shot learning are significant non-trivial contributions.

The paper is well written, and draws from and builds upon the current literature on Latent ODE models. The paper shows extensive empirical analysis on multiple datasets and compares the proposed framework to modern baselines. The authors have provided code and sufficient details in the paper to reproduce the analyses.

**Q4 Main Weakness:**

There are no major weaknesses to the paper.


**Q5 Detailed Comments To The Authors:**

I don't have any major feedback. The paper looks good in its current form. A minor suggestion is to include a more detailed figure in the paper to describe how all the different pieces of the model fit together. It would be easier to get an overall idea of the framework if Figure 1 could be made more descriptive or if Algorithm 1 from supplementary material could be moved to the main paper.

**Q7 Justification For Your Score:**

The paper is technically strong and has some novel ideas in terms of combining features, uncertainty quantification, and zero-shot learning. The ideas are presented clearly with enough details to reproduce the results.

**Q9 Complying With Reviewing Instructions:**

1: Yes.

---

### Decision · Program_Chairs · 2022-05-15

**Decision:**

Accept (Poster)

**Comment:**

Meta Review: This paper proposes a learning framework for time-series data where a ODE model in a latent space is learned together with a "measurement process" from a series of system "inputs" and "outputs". This is a generative model, which allows natural interpolation of time series and prediction of output for new inputs.

Pros:

Reviewers liked the extensive use of real (biological) data to evaluate and illustrate the approach. Most but not all reviewers felt the framework was interesting and potentially innovative due to its generality and flexibility; these properties could allow the approach to be used in quite diverse practical applications.

Cons:

The main criticism was that the learned latent space representation does not necessarily have any mechanistic meaning, i.e., there is no guarantee that one will identify variables that represent or can be mapped to the actual mechanistic actors that generate the observed data. In their response, the authors state that they do not claim the latent space to have mechanistic meaning. However, the paper does contain several sentences that include the word "mechanistic" and could be read in this way to create this potential misunderstanding of the authors' claims.

Upon consultation with the reviewers, the consensus was that this issue does not necessitate rejecting the paper, and a majority was in favor of acceptance with one reviewer stating that they still did not find the paper convincing but would not mind. The overall opinion is therefore best described as a "weak accept". If accepted, the authors should be strongly encouraged to carefully revise their paper's wording in order to avoid being seen as making mechanistic claims.